# Annual-to-millennial fluctuations in the physical properties of crystal-rich magma storage zones

Oliver Higgins [1,2] ✉, Michael J. Stock[2], Dennis Geist[3], David A. Neave [4], Iris Buisman[5], Benjamin Bernard [6] & Matthew Gleeson [7]

Basaltic melts may variably disaggregate macrocrysts (large crystals) from crystal mushes during the assembly of magma bodies beneath ocean-island volcanoes. The entrained macrocrysts modulate the crystallinity, density, and rheology of magmas, parameters that control magma system architecture and eruptive dynamics. However, the timescales, drivers, and consequences of inconstant crystal mush incorporation into carrier melts require quantification. Here, we use a suite of plagioclase-rich basalts to show that the entrainment efficiency (the ability for melts to disaggregate and entrain macrocrysts from crystal mushes) is temporally variable on inter-eruption timescales at ocean-island volcanoes. Macrocryst cargoes are predominantly out of equilibrium with their carrier melts in both chemistries and mass proportions (ratios of different macrocryst phases). Geochemical and petrological evidence reveals that macrocryst mass proportions are established in a density-stratified melt-rich reservoir shortly before eruption, whereas the absolute crystallinity is a function of crustal physics, likely driven by fluctuations of annual-to-millennial melt supply. Variations in entrainment efficiency explain several universal, but enigmatic, features in oceanic volcanic systems, such as decoupled crystallinity–temperature–time relationships and the dearth of plagioclase-rich basalts at fast-spreading mid-ocean ridges. Systematically studying temporally constrained eruptive products offers a unique window into the evolution of crystal-rich magma storage zones.

Petrological, geochemical, seismological and field-based investigations present magmatic systems underlying ocean-island volcanoes as vertically extensive stacks of interconnected sills[1–3]. However, the spatiotemporal evolution of these complex magmatic systems (e.g. rheology, mechanical strength of the crust, melt focusing) and the impact on magma assembly and eruptive dynamics remain enigmatic. Unlike the magmatic systems beneath mid-ocean ridges (MOR) and subduction zones, the intrusive portions of ocean islands are rarely, if ever, exposed. Therefore, their geometries and geochemical evolution are typically deduced from studying erupted products[4,5], crustal xenoliths[6,7], and geophysics[8] alone. A common observation in ocean-island settings is that the mass fraction of macrocrysts (large [mm-cm], low-aspect-ratio crystals, texturally distinct from fine-grained, high-aspect-ratio groundmass) may vary between consecutive eruptions on annual-to-millennial timescales[9–11]. Changes to magma crystallinity may reflect: (i) spatiotemporal fluctuations in the physical properties of the crust which in turn affect the susceptibility of pre-existing crystalline material (e.g. crystal mush) to be disaggregated, transported, and erupted by ascending carrier melts (hereafter the entrainment efficiency)[12]; or (ii) the role of intensive variables (e.g. temperature, pressure, melt water content, oxygen fugacity) during magma storage and in situ equilibrium crystallisation[13]. Therefore, unlocking the origin of magmas' diverse crystallinities may offer unique insights into the physical-chemical structure of sub-volcanic magmatic systems.

Interpreting whether magma crystallinity is an extrinsic property of the crust, such as the physical structure of crystal mush, or an intrinsic property of magmas through in situ crystallisation conditions requires consideration of mineral–melt equilibria. If macrocrysts grow incrementally from a melt under equilibrium conditions, then crystallinity is negatively correlated with magma temperature[14]. However, textural-chemical studies often conclude that most macrocryst cores are mechanically scavenged from crystal mushes

[1]School of Earth and Environmental Sciences, University of St Andrews, St Andrews, UK. [2]Discipline of Geology, School of Natural Sciences, Trinity College Dublin, Dublin, Ireland. [3]Geology Department, Colgate University, Hamilton, NY, USA. [4]Department of Earth and Environmental Sciences, The University of Manchester, Manchester, UK. [5]Department of Earth Sciences, University of Cambridge, Cambridge, UK. [6]Instituto Geofísico, Escuela Politécnica Nacional, Quito, Ecuador. [7]Department of Earth and Planetary Science, University of California, Berkeley, CA, USA. ✉e-mail: ojh4@st-andrews.ac.uk

by their carrier melts shortly before eruption rather than grown in situ[12,15,16]. This raises the possibility that crystallinity and pre-eruptive storage temperature of magmas are fundamentally decoupled. First, an exotic macrocryst origin is inferred for many basaltic magmas erupting with mass proportions (ratios) of macrocryst phases discordant with equilibrium crystallisation experiments[15,17,18]. Plagioclase-rich basalts, which are over-enriched in plagioclase (e.g. mass proportions of plagioclase >0.8[17]) with sparse clinopyroxene and olivine[10,19], epitomise this feature. Second, macrocryst cores may be out of equilibrium with their carrier melts, with crystals from a single magma exhibiting diverse chemical zoning patterns[16,20,21] and isotopic compositions[22]. Disequilibrium features in volcanic rocks are mirrored in intrusive suites such as ophiolite sequences, layered igneous intrusions, and erupted cumulate nodules[23–25]. Similarities include disequilibrium modal layering in the form of anorthosites or leuco-gabbros paired with complementary wehrlites (clinopyroxene + olivine; e.g. Rum in Scotland[26], Skaergaard in Greenland[27] and Stillwater in the USA[24]), as well as distinctly zoned macrocrysts juxtaposed over short length scales[28].

Here, we use a series of plagioclase-rich basaltic magmas from Volcán Wolf (Isabela Island, Galápagos Archipelago) with temporally variable macrocryst cargoes to demonstrate that the entrainment efficiency is a fundamental and highly changeable parameter that determines magma crystallinity. Furthermore, we show that entrainment efficiency varies on annual-to-millennial timescales at other mafic volcanoes, where comparable crystallinity data are reported, and can control the rheology of basaltic magmas and so their eruptive behaviour.

## Results and discussion
### The chemical signature of macrocryst accumulation in basalts
The Galápagos Archipelago comprises an active ocean-island volcanic system located ~1000 km west of coastal Ecuador and represents the surface expression of the Galápagos mantle plume (Supplementary Fig. 1). Collectively, Western Galápagos volcanoes are excellent natural laboratories for investigating the physico-chemical properties of ocean-island magmatic systems because of their frequent eruptions[29], well-constrained geophysics[8,30,31] and thermobarometry[32], and petrographically diverse basalts[10,33,34]. Volcán Wolf (Isabela Island, Western Galápagos) has been subject to several field campaigns in the last few decades, providing access to fresh basaltic lava flow samples that cover a range of vent geometries as well as a chronology established by both absolute ($^{40}Ar/^{39}Ar$, $^3He$) and historical

age constraints[8,34,35]. Figure 1a shows the bulk geochemistry of erupted products from Volcán Wolf in MgO–$Al_2O_3$ space, in which data form two trends. At <14.7 wt% $Al_2O_3$, MgO decreases with $Al_2O_3$, with these low-Al samples including all lavas and matrix glasses from the well-studied 2015 eruption[8,32]. At ≥14.7 wt% $Al_2O_3$, the $Al_2O_3$ increases across a limited span of MgO, with these high-Al samples plotting colinear to the array of vectors projecting to plagioclase mineral compositions in MgO–$Al_2O_3$ space (i.e. mineral mixing lines; red triangular field in Fig. 1a). Furthermore, the plagioclase-compatible trace element[36] Sr increases systematically with $Al_2O_3$ in high-Al samples (Fig. 1a).

Geochemically, high-Al magmas bear the hallmarks of variable plagioclase macrocryst accumulation. To independently confirm this, we collected a series of thin-section scale chemical maps across a range of whole-rock compositions (large symbols in Fig. 1a; $n = 32$) to calculate the crystallinity and mass proportion of macrocryst phases (plagioclase, olivine, clinopyroxene; see 'Methods'). Macrocryst mass proportions (Fig. 1b) are dominated by plagioclase (0.90 ± 0.22; median ± inter-quartile range) and sparse in both clinopyroxene (0.07 ± 0.12) and olivine (0 ± 0.04) across the full range of magma crystallinity (>0–25 wt%). Volcán Wolf macrocryst mass proportions are akin to plagioclase-rich basalts[10,15,19] globally; based on experiments that employ bulk compositions similar to Volcán Wolf magmas[37,38], they are far too rich in plagioclase to be equilibrium mineral assemblages (Fig. 1b). Furthermore, plagioclase is not the sole liquidus phase under any pressure–temperature–composition conditions in these experiments. Hence, as anticipated from Fig. 1a, most of the variance in whole-rock $Al_2O_3$ is generated by varying the magma crystallinity, predominantly by changing the amount of plagioclase. Low-Al magmas have low crystallinities (0–3 wt%) and so may be considered natural melts alongside low-Al tephra-hosted matrix glass collected from the 2015 eruption[8]. Crystallinities increase up to 25 wt% in high-Al magmas (Fig. 1b; Supplementary Fig. 2), which cannot be considered true melt compositions.

To determine whether high-Al magmas are mechanical mixtures of a low-Al, macrocryst-poor carrier melt and a leuco-gabbroic (plagioclase-rich) macrocryst assemblage, as suggested by Fig. 1a, we used a chemical mass-balance approach (see 'Methods'). In short, the solid macrocryst assemblages (plagioclase ± clinopyroxene ± olivine), as determined from chemical mapping, were subtracted from their corresponding high-Al whole-rock compositions to calculate the compositions of the theoretical carrier melts. The calculated carrier melts were then compared to erupted

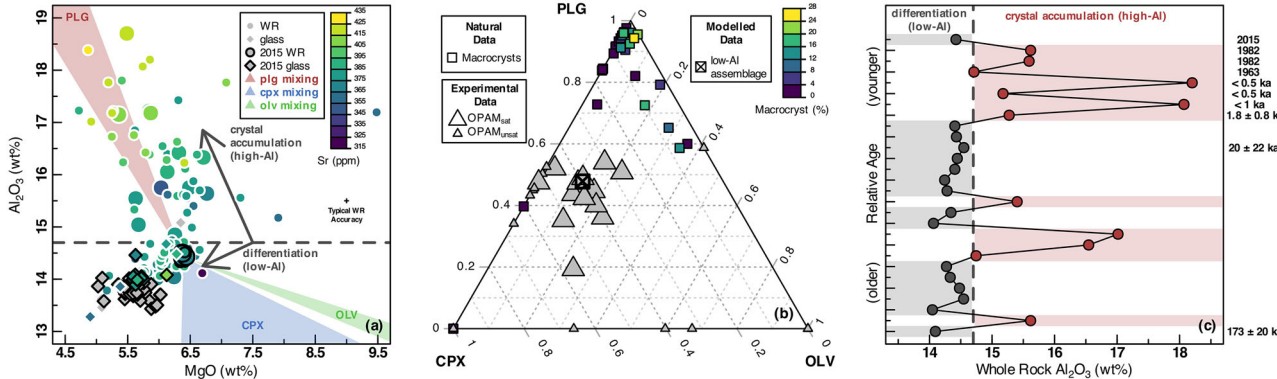

**Fig. 1 | Geochemistry and petrology of Volcán Wolf erupted products. a** $Al_2O_3$ (wt.%) versus MgO (wt.%) of Volcán Wolf magmas (circles) and matrix glasses (diamonds), colour contoured for bulk Sr (ppm) where available. Coloured fields show the family of mixing lines projecting towards chemical analyses of plagioclase (plg; red), clinopyroxene (cpx; blue), and olivine (olv; green) measured for Volcán Wolf. Whole-rock chemistry diverges at ~14.7 wt% $Al_2O_3$, below which magmas evolve by simple gabbroic fractionation (low-Al trend) and above which magmas lie coincident to the plagioclase mixing field (high-Al trend). Large symbols are samples selected for thin-section scale chemical mapping. Data sources are[8,34,35,68–70] and this study. Error bar is relative percentage error (accuracy) based on repeat analyses

($n = 4$) for standard BHVO-1[56]. **b** Ternary diagram of plagioclase, clinopyroxene, and olivine macrocryst mass proportions for chemically mapped natural samples and selected equilibrium experiments[37,38]. Equilibrium experiments are divided between OPAM-saturated and OPAM-unsaturated melts, neither of which reproduces Volcán Wolf macrocryst proportions. The best-fit proportions of minerals that drive the evolution of low-Al melts by equilibrium crystallisation is shown as a cross. **c** A sequence of Volcán Wolf lavas from Geist et al.[35] with relative temporal (stratigraphic) constraints and some absolute dating. Data are coloured for magmas above (red) and below (grey) the trend-discriminating $Al_2O_3$ (14.7 wt%) observed in (**a**). High-Al and low-Al eruptive periods alternate through relative time.

**Fig. 2 | Molar mineral chemistry of mafic minerals in Volcán Wolf magmas. a** Plagioclase [XAn], **b** clinopyroxene [XMg], and **c** olivine [XFo]. Calculated mineral–melt equilibria indicate that macrocrysts are predominantly out of equilibrium with low-Al carrier melts. Mineral compositions from equilibrium experiments relevant to Volcán Wolf[37,38] are shown at the top of each panel. For plagioclase (**a**), published data from mid-ocean ridge (MOR)[15] and ocean-island[9,47] basalts are shown for comparison; red violins are measured XAn from plagioclase macrocrysts and grey violins are the predicted equilibrium XAn[20] for carrier melts.

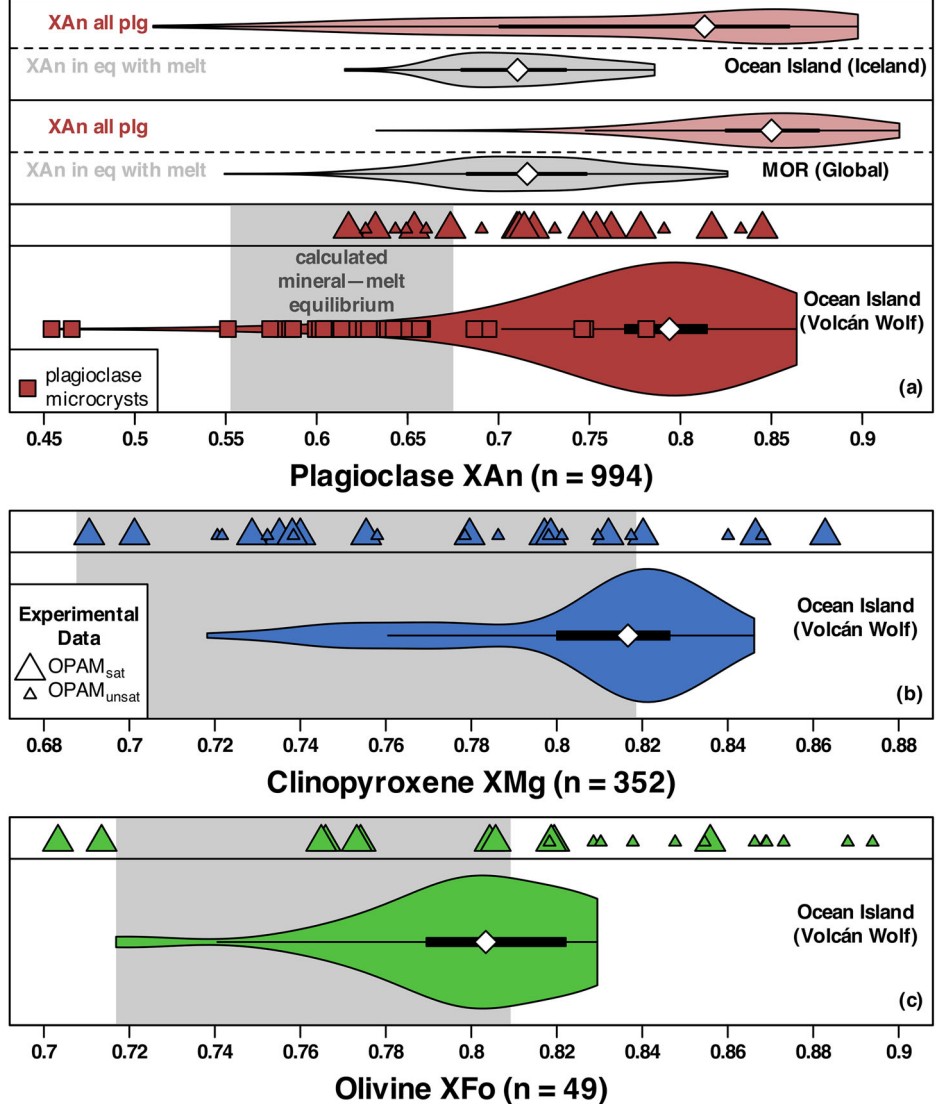

Volcán Wolf magmas (Supplementary Fig. 3). Despite uncertainties implemented by the span of macrocryst chemistries, high-Al magmas can indeed be explained by a simple mixing scenario. The calculated liquids are coincident to the liquid line of descent elucidated by low-Al magmas (e.g. Fig. 1a). We also note that high-Al and low-Al magmas are distributed in packages through time, with stratigraphic periods in which one magma type is more likely to be produced than the other (Fig. 1c). The 2015 eruption is a case in point: no high-Al basalts were identified[8], whereas other recent historical eruptions produced only high-Al basalts (Fig. 1c).

Most macrocryst compositions from Volcán Wolf are out of major-element equilibrium with any material identified as a carrier melt (e.g. low-Al matrix glasses or low-Al magmas; Fig. 1a), according to chemistry-based tests (see 'Methods'). Specifically, plagioclase with XAn < 0.55 or XAn > 0.67 (95% of analyses; Fig. 2a; XAn = molar Ca/[Ca + Na + K]), clinopyroxene with XMg > 0.82 (47% of analyses; Fig. 2b; XMg = molar Mg/[Mg + Fe]), and olivine with XFo > 0.81 (31% of analyses; Fig. 2c; XFo = molar Mg/[Mg + Fe]) are not equilibrium mineral compositions. Many Volcán Wolf plagioclase macrocrysts have diverse zoning patterns, both within and between samples, although transects tend to converge at a common low-XAn rim (0.55–0.67) which is in equilibrium with low-Al carrier melts (Supplementary Fig. 4) and overlaps with groundmass plagioclase chemistry (Fig. 2a). In general, the disequilibrium between basaltic carrier melts and macrocryst cores, best shown and most often reported for the mineral

plagioclase, is a common feature of global ocean-island and MOR basalts[9,15,18,20,21] (Fig. 2a).

In summary, our petrographic and geochemical observations provide several arguments against in situ crystallisation as the driver of the chemical differences between the high-Al magmas of Volcán Wolf: (i) high-Al magmas fall along a mixing line between low-Al magmas and plagioclase, and not a typical plagioclase-saturated basaltic liquid line of descent (where Al and Mg would be positively correlated[39]) as is the case for the low-Al magmas (Fig. 1a); (ii) a lack of experimental analogues where plagioclase could crystallise in such consistently high (0.90 ± 0.22; median ± interquartile range) mass proportions (Fig. 1b); (iii) a strong disequilibrium between macrocryst cargoes and natural melts, the latter represented by low-Al matrix glasses and low-Al magmas (Fig. 2); and (iv) no relationship between plagioclase macrocryst compositions and whole-rock $Al_2O_3$ (Supplementary Fig. 4).

## Magma crystallinity is a function of survivability and entrainment efficiency

The diverse basalt geochemistry of Volcán Wolf reveals fundamental mechanisms which modulate crystallinity in the erupted products of the oceanic crust over annual-to-millennial timescales (Figs. 1c and 3). The plagioclase-rich macrocryst mass proportions (i.e. the ratio of different macrocryst phases to one another; Fig. 1b) are most simply explained by

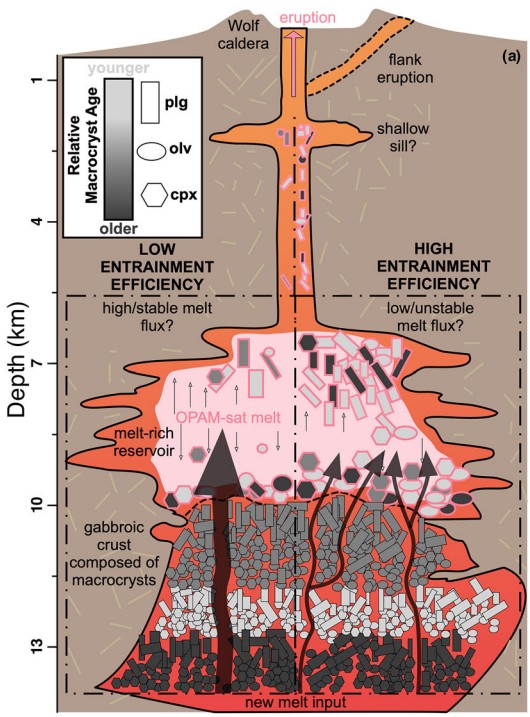

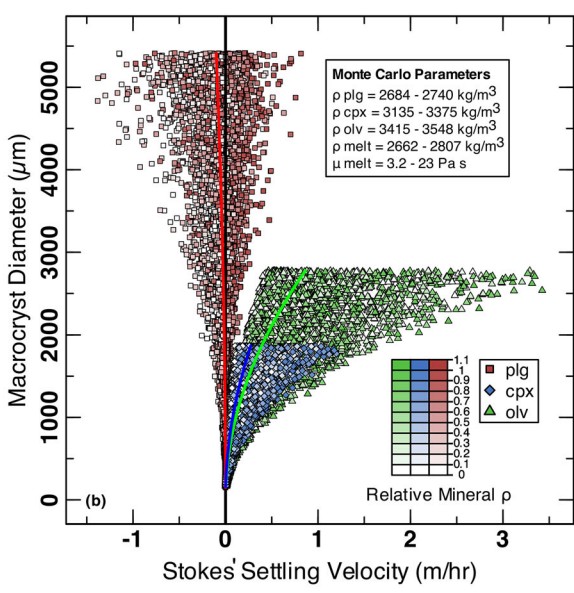

**Fig. 3 | Model depicting magma assembly as a function of macrocryst surviva-bility and entrainment efficiency. a** Genetic diagram indicating the model by which macrocryst-poor (left side) and macrocryst-rich (right side) magmas are assembled. **b** Macrocryst diameter (μm) versus Stokes' settling velocity (m/hr) for plagioclase, clinopyroxene, and olivine. Stokes' settling velocity is calculated via a Monte Carlo method ($n = 10,000$; sampling uniformly within parameter bounds with replace-ment; see 'Methods'). Coloured lines represent averages for each mineral phase as a function of macrocryst diameter. Note that plagioclase is efficiently fractionated even at small macrocryst diameters.

density-driven phase separation in a melt-rich reservoir during pre-eruptive magma storage (see 'Methods' for parameter derivations and density/viscosity calculations). For the range of measured macrocryst chemistries, plagioclase (2684–2740 kg/m³) is always less dense than clinopyroxene (3135–3375 kg/m³) or olivine (3415–3548 kg/m³). Considering the range of melt viscosities (3.2–23 Pa s), melt densities (2662–2807 kg/m³), and macrocryst diameters, Stokes' law dictates that plagioclase is always at or close to neutral buoyancy in its carrier melt whereas olivine and clinopyroxene sink (Fig. 3b). Although hindered settling may reduce settling velocities by >60%[40], with additional reduction by convection or pressure-driven flow, density-driven settling in a melt-rich reservoir is the most viable method to consistently generate disequilibrium mineral mass proportions. Dynami-cally, plagioclase has a high survivability in the melt, forming a passive mineralogical tracer, whereas clinopyroxene and olivine will rapidly form a basal wehrlite cumulate that is isolated from the eruptible system.

The depth of the melt-rich reservoir, and so the locus of the density-driven separation process, can be approximated using thermobarometry. As with many basaltic volcanoes, Volcán Wolf erupted melts are multiply saturated at the three-phase gabbro cotectic (Olivine + Plagioclase + Augitic clinopyroxene + Melt; OPAM_sat)[8,32]. The composition of OPAM_sat melts is pressure- and temperature-dependent[41,42], facilitating melt-only thermobarometry. Using the model of Higgins and Stock[32] applied to low-Al carrier melts, we find a universally strong probability of OPAM saturation (0.90 ± 0.02; median ± inter-quartile range) at uniform pressure (2.6 ± 0.5 kbar) and temperature (1162 ± 21 °C) conditions, despite the range of erupted magma compositions (Supplementary Figs. 5 and 6). This calculated pressure is in excellent agreement with pre- and syn-eruptive interferometric synthetic-aperture radar (InSAR) inversions for the 2015 Volcán Wolf eruption[8], indicating that low-Al melts equilibrated in a mid-crustal (8.9 ± 1.7 km; pressure–depth conversion from Higgins and Stock[32]) reservoir immediately prior to eruption. As parts of macrocrysts (e.g. the outermost plagioclase crystal rims) are in equilibrium with low-Al carrier melts (Fig. 2), which themselves equilibrated at an average of 2.6 kbar,

further processing at shallower crustal levels or at the surface is unlikely. The narrow range of pressure suggests that the secondary, shallow (<2 km) reservoir beneath the caldera identified by InSAR[8] may not store large volumes of eruptible magma. The average mass proportion of olivine, pla-gioclase, and augitic clinopyroxene driving the low-Al trend is 0.12, 0.48, and 0.40, respectively. These calculated proportions differ from the observed plagioclase-dominated mass proportions of accumulated macrocrysts in the magmas (Fig. 1b), instead representing the final OPAM_sat crystallisation of macrocryst rims and matrix crystals in equilibrium with the low-Al carrier melt prior to eruption (Fig. 3a). They are, however, concordant with mineral mass proportions in OPAM_sat experiments[37,38] (Fig. 1b).

We have shown that disequilibrium macrocryst mass proportions are explained by pre-eruptive, density-driven settling as described above (Fig. 3). However, as the temperature of the carrier melt and the macrocryst cargo are decoupled, there is no viable way to alter the crystallinity of consecutive eruptions within the melt-rich reservoir. Instead, our data are consistent with a changing entrainment efficiency of crystal mushes from the lower crust prior to final pre-eruptive storage in a mid-crustal magma reservoir that acts as a density filter for the macrocryst cargo (Fig. 3a). In this framework, gabbroic crystal mush is physically disaggregated by an ascending carrier melt at or below the level of final mid-crustal storage (2.6 ± 0.5 kbar; 8.9 ± 1.7 km) and equilibration. In basaltic systems, mechanical disaggregation is favoured over the remelting germane to silicic systems due to the smaller temperature contrast between carrier melts and crystal mush in the former[43]. The original mass proportion of disaggregated and entrained mineral phases from the crystal mush is erased by late-stage density separation in a mid-crustal melt-rich reservoir. Potentially, the accumulated proportions resembled the cluster of OPAM mineral pro-portions from equilibrium experiments in Fig. 1b, which yield macrocryst chemistries spanning the entire Volcán Wolf array (Fig. 2). For reference, clinopyroxene thermobarometry records depths which overlap with the ~2.6 kbar locus of melt storage but also reach greater depths[8,32], inferring that clinopyroxene may have been entrained from the deeper crust.

Therefore, the range in magma crystallinity (up to 25 wt%; Fig. 1b) is primarily generated by varying the ratio of entrained gabbroic mush to carrier liquid through time.

The Volcán Wolf basalts provide compelling evidence that crystallinity and temperature are decoupled and that crystallinities are likely set in the lower to middle crust. The crystallinity is controlled by varying the entrainment efficiency prior to the emplacement of a magma-melt slurry as a liquid-rich mid-crustal reservoir (Fig. 3a), whereas the macrocryst mass proportions are established during a subsequent density-driven phase separation process prior to eruption (Fig. 3b). The erupted end product is therefore a function of macrocryst survivability in a melt, which favours plagioclase-rich assemblages, and the entrainment efficiency of pre-existing crystalline material from the deeper oceanic crust. By extension, the physical properties of magmas, which are intrinsically connected to crystallinity and control eruptive dynamics, such as density and effective viscosity[44], are determined at or below mid-crustal depths.

## Spatiotemporal evolution of entrainment efficiency: mechanisms and implications

Our physico-chemical model (Fig. 3a), featuring a melt-rich reservoir in the middle crust that is supplied by magmas of varying crystallinities on annual-to-millennial timescales, is widely supported both regionally and globally in ocean-island settings. InSAR deformation measurements from western Galápagos volcanoes are explained by an elongate reservoir on the order of 10s–100s of metres thick[8,31]. Plagioclase-rich enclaves from Rabida have macrocryst chemistries and textures suggestive of aggregation in a melt-rich environment[45] whereas complementary wehrlite residua are evinced by cumulate fragments in the Galápagos[6] as well as the Canary Islands[23,46]. Short storage timescales in a mid-crustal reservoir (months or years) are consistent with the thin rims on accumulated plagioclase macrocrysts, another globally noted feature in ocean-island basalts[9,10,17-19,47,48]. The density and viscosity ranges for Volcán Wolf melts and macrocrysts, used as physical parameters in our simple Stokes' settling model (Fig. 3b), are equivalent to those of tholeiitic magmas erupted at global ocean islands as well as MORs[15,17], which exemplifies the ubiquity of this process for generating disequilibrium macrocryst mass proportions.

A fundamental observation is that entrainment efficiency drives differences in crystallinity on annual-to-millennial timescales (Fig. 1c). Although entrainment efficiency could be a function of the compactness of the crystal mush framework[11], solidification of mush in the deep crust occurs on timescales many times longer than repose between eruptions with widely different cargoes. Instead, we propose that the melt supply or the repose between individual melt pulses is the limiting factor[49]. Where melt supply is higher or more established, channelised flows through the gabbroic crust can develop, forging stable ascent pathways and favouring aphyric or crystal-poor magma production[50]; channelisation occurs on far shorter timescales than the modulation of long-term melt flux by fluctuating plume temperatures or changing distance from the plume in ocean islands[6,51]. At MORs, tectonic settings which share petrographic similarities with ocean islands, a relationship between entrainment efficiency and melt supply is also supported. In the MOR environment, parameters such as melt Mg content, mantle compositional variance, and time-averaged melt supply correlate with spreading rate[52]. However, plagioclase-rich basalts are not observed in fast-spreading ridges but are common at slow- and intermediate-spreading ridges[15]; the dearth of crystal-rich magmas has previously been linked to the filtering effect of a permanent axial melt lens[52], although we proffer that the establishment of a focused and efficient melt supply reduces the entrainment efficiency and so the crystallinity.

Volcán Wolf data suggest that entrainment efficiency is more likely to be temporally, rather than spatially, variable. Whole-rock $Al_2O_3$, which acts as a proxy for crystallinity, changes through time (Fig. 1c), yet there is no relationship with vent location or elevation (Supplementary Fig. 7). Similar temporal variations in crystallinity are observed in a variety of mafic volcanic settings. Magmas of the Bárðarbunga-Veiðivötn volcanic system (Iceland), spanning Early Holocene to historical eruptions, show large fluctuations in crystallinity (5–45 vol%) through time[11]. Neogene lava flows of the Grænavatn porphyritic group from eastern Iceland also exhibit alternating periods of plagioclase-rich and aphyric lavas, with plagioclase abundance varying more between flows than within flows[9]. However, sequential eruptions from Stromboli (Italy)[53] and Fernandina (Western Galápagos)[33] show that changes in entrainment efficiency may also occur on yearly, as opposed to millennial, timescales, in agreement with our data from Volcán Wolf (Fig. 1c).

Our observations have implications for both intrusive and extrusive oceanic magmatic systems. In extrusive volcanic systems, the link between entrainment efficiency and magma crystallinity may fundamentally influence eruptive dynamics. For effusive eruptions, the addition of macrocrysts can cause a transition to a non-Newtonian shear thinning rheology, enabling crystal-rich flows to rapidly advance during periods of high effusion rate[9]. For explosive eruptions, crystallinity may strongly influence effective viscosity[44], which has feedbacks with eruptive style[54]. In intrusive systems, fluctuating entrainment efficiency may generate the repetitive modal layering common to mafic igneous intrusions when injection of a crystal-laden carrier melt into the crust is followed by magma stalling rather than eruption: (i) a mid-crustal reservoir is established as shown in Fig. 3a but does not erupt; (ii) cycles of magma injection followed by density separation establishes layers with non-cotectic mineral proportions of variable thickness due to a temporally variable entrainment efficiency; and (iii) slow (percolative?) loss of melt into the overriding crust leaves behind the cap of plagioclase residue, with some in situ crystallisation of interstitial melt when increasing crystallinity causes viscous jamming.

At present, the coarse resolution of geophysical imaging techniques precludes real-time observations of the spatiotemporal processes operating in the oceanic crust on sub-kilometre length scales. Hence, the evolution of the physical properties of the crust is only elucidated using indirect petrological methods; chemical-textural studies of magmas and intrusive suites can uniquely trace macrocryst entrainment efficiency for the entire erupted history of an active magmatic system.

## Methods
### Sample selection and chronology
Volcán Wolf (91.35 °W, 0.02 °N) is a 1710 m shield volcano on the northernmost tip of Isabela Island. Morphologically, it has the same "inverted soup bowl" form as the other volcanoes in the western Galápagos, covered by three spatial clusters of vents: lower, radially distributed flank fissures; arcuate fissures with an alignment sub-parallel to the caldera wall; caldera floor vents[35]. Steep upper slopes and voluminous eruptions from the radial vents both promote 'A'ā lava flows, with only small-volume pahoehoe flows on the plateau between the caldera wall and the steep outer flanks[35]. All samples forming the stratigraphy outlined in Fig. 1c are from a 1995 field campaign by D. Geist[35], mostly through sampling the stratigraphy of an exposed section of caldera wall. Attempts to date the erupted lavas across the volcanic edifice have proved difficult due to propylitic alteration of the lowermost flows, yet some attempts show that the oldest dateable lava flow has a $^{40}Ar/^{39}Ar$ plateau age of $173 \pm 20$ ka and thus the main episode of volcano shield growth was well-established by 100 ka[35]. Furthermore, 5 flank lavas yielded ages of $1800 \pm 800$ years to <500 years which suggests that any flank lavas discussed in this study are likely to have erupted in the last few thousand years. A clear relative chronology is also observed using satellite imagery, where lava flows have spectral signatures that show a proportionality between vegetation cover and eruption age. These relative and absolute chronological observations help to establish that crystallinity is changing on yearly to millennial timescales for Volcán Wolf. All chemically mapped samples are from the field campaign of M. Stock, M. Gleeson, and B. Bernard in June 2017. The 2015 eruption samples have been studied extensively[8,32,34], whereas the older lava flows (comparable to the samples from Geist et al.[35]) are discussed for the first time here.

### X-ray fluorescence analysis of whole-rock samples
Volcán Wolf whole-rock data used in this study are predominantly from three sources: (i) new, unpublished, data from various low-vesicularity lava

flows of unspecified ages (this study; sampled June 2017); (ii) lava flow samples of the 2015 eruption, all recovered from flow lobes on the eastern flank of Volcán Wolf and covering the circumferential fissure phase of the eruption[34]; and (iii) samples from a 1995 field campaign which include a mixture of flank and circumferential fissure lava flows[35]. All whole-rock data, both new and previously published, are presented in Supplementary Data 1a. The data used in figures and calculations are normalised to 100 wt% anhydrous, with all Fe as FeO, unless otherwise stated. Melt cation fractions, where used for calculations or modelling, were calculated as per Putirka[55] (see their Table 1 for a worked example). The sampling locations are plotted in Supplementary Fig. 1 and a range of Harker variation plots are presented in Supplementary Fig. 6.

Whole-rock samples from this study and Stock et al.[34] were analysed by X-ray fluorescence spectrometry (XRF) for major and trace elements via glass discs and pressed powder pellets, respectively. XRF analyses were performed with a Philips PW 2404 instrument at the University of Edinburgh. Data precision, including uncertainties associated with sample preparation and whole-rock heterogeneity, was estimated by analysing three replicates of the same natural sample. Three to four replicate analyses of secondary standards[56] were used for quality control in both major- and trace-element analyses: BHVO-1 for all major elements, BHVO-1, BCR, and BIR for Zn, Cu, Ni, Cr, V, Ba, Sc; BHVO-1, BCR, and JB2 for U, Th, and Pb; BHVO-1, BCR, and BIR-1 for Nb, Zr, Y, Sr, Rb. The accuracy of major- and trace-element concentrations was calculated using the mean value from the repeat standard analyses and expressed in percent relative (Supplementary Data 1b).

Major and selected trace elements in whole-rock samples from Geist et al.[35] and three of their unreported samples, now presented in this study, were measured by XRF at Washington State University. Analyses were performed according to established techniques[57] who also report analyses of international standards. Rare-earth elements were measured by Geist et al.[35] on a subset of samples at Lawrence University by inductively coupled plasma mass spectrometry (ICP-MS). Analytical details for the ICP-MS analyses are provided elsewhere[58].

### Electron probe microanalyser (EPMA) of tephra matrix glass and silicate minerals

For interpretation of whole-rock data, and for use in our modelling (see below), we used published EPMA analyses of tephra matrix glass and clinopyroxene[8] as well as plagioclase and olivine[34]. Tephra samples were collected from seven locations on the east coast of Volcán Wolf[8]. No tephra was found on top of any 2015 lava flows, including the initial southeast flow lobe, and so we follow the interpretation that it was expelled during the high lava fountaining episode at the onset of eruption[59]. All mineral and tephra matrix glass analyses are from 2015 erupted products and were analysed using a Cameca SX100 EPMA in the Department of Earth Sciences at the University of Cambridge. Full EPMA analysis conditions and uncertainties are reported elsewhere[8,34]. Mineral formula recalculations were performed for olivine, plagioclase, and clinopyroxene on the basis of four, eight, and six oxygens, respectively. The chemical components in clinopyroxene were also calculated using two similar methods[55,60].

We performed additional EPMA analyses for plagioclase on 6 samples from this study to verify that plagioclase mineral chemistry did not vary as a function of whole-rock chemistry or bulk crystallinity (Supplementary Fig. 4). Analyses were performed on a JEOL 8200 Superprobe in the Department of Archaeology at the University of Oxford. Analysis conditions were 15 keV, 5 μm beam, and 40 nA, using transects from rim to core to maintain positional sensitivity. The raw output file from the EPMA, which includes peak positions, backgrounds, secondary standards, and setup, can be found in Supplementary Data 2. Note that plagioclase chemistry from this study overlaps precisely with plagioclase from the 2015 eruption[34], indicating that plagioclase mineral chemistry shows no relationship with eruption, bulk chemistry, or crystallinity. This finding further supports our hypothesis of a mechanical disaggregation of gabbroic crustal material into a basaltic carrier melt.

### Mineral chemistry and equilibria

The mineral chemistry of plagioclase (XAn; molar Ca/[Ca + Na + K]), olivine (XFo; molar Mg/[Mg + Fe]), and clinopyroxene (XMg; molar Mg/[Mg + Fe]) is summarised in Fig. 2. The range of equilibrium mineral compositions is calculated using existing mineral–melt equilibria models applied to tephra matrix glass and whole rock with <14.7 wt% $Al_2O_3$ (low-Al samples), which are updated from the previously published values[8,34]. For olivine, we consider a $Kd^{Fe-Mg}$ of 0.299 ± 0.053 as equilibrium[55]. For plagioclase, we used a model for low-$H_2O$ basaltic melts[20]. For clinopyroxene, we used a model which predicts the EnFs (enstatite + ferrosilite), DiHd (diopside + hedenbergite), and CaTs (calcium tschermak) components in equilibrium with melt at a given pressure, temperature, and melt $H_2O$ content[61]. We used the temperature calculated from Eq. 33 of Putirka et al.[55], a representative mean pressure of 2.6 kbar from OPAM barometry[32], and a melt $H_2O$ content of 0.7 wt%. A clinopyroxene is deemed in equilibrium with its melt pair if the predicted and measured EnFs, DiHd, and CaTs components are within ±0.05, ±0.06, and ±0.03, respectively[38,61] and the $Kd^{Fe-Mg}$ is within ±0.03 of that predicted by Eq. 35 of Putirka et al.[55].

### Calculation of intensive parameters (pressure, temperature, density, viscosity, Stokes' settling velocity)

Pressure and temperature of Volcán Wolf liquids (low-Al crystal-poor whole rocks and matrix glasses) were calculated using the OPAM thermobarometer for liquids multiply saturated in a gabbroic assemblage of olivine + plagioclase + augitic clinopyroxene[32]. The OPAM thermobarometer also returns a chemical probability of OPAM saturation which is much greater than 0.5 for most of the tested melt compositions (low-Al samples) from Volcán Wolf (Supplementary Fig. 5). The high chemical probability of OPAM saturation is in good agreement with petrographic observations given that all samples contain the three OPAM mineral phases and, with the exception of sparse Fe-oxides reaching 10s of μm in length, no other mineral phases were identified. OPAM thermobarometry for 2015 erupted products from Volcán Wolf have been previously reported[32]. The pressures and temperatures of additional historic liquids calculated in this study show no observable difference from these 2015 samples, despite covering a slightly wider compositional range (Supplementary Fig. 6).

The density (Supplementary Data 3) of Volcán Wolf liquids was calculated using the DensityX online interface[62] and the viscosity (Supplementary Data 3) using a widely applied composition- and temperature-dependent model[63]. The Stokes' settling velocity was calculated using a Monte Carlo approach ($n = 10,000$), by taking bounds for all relevant parameters and randomly sampling according to a uniform distribution. The following bounds were used:

– Melt density: the minimum and maximum of calculated melt densities from DensityX[62].
– Melt viscosity: the minimum and maximum of calculated melt viscosity[63].
– Melt $H_2O$ content: 0–0.7 wt%
– Plagioclase, clinopyroxene, and olivine density: minimum and maximum densities using the range of EPMA analyses of Volcán Wolf mineral and end-member densities (anorthite, albite, forsterite, fayalite, wollastonite, enstatite, ferrosilite)[64].
– Plagioclase, clinopyroxene, and olivine crystal diameters: minimum and maximum diameter of macrocrysts as measured from QEMSCAN maps for the whole sample suite.

### Major-element mass balance modelling

Lever rule mass balance calculations were established to assess if a crystal-rich, high-Al magma could be chemically explained as a mechanical mixture of a low-Al carrier melt and a macrocryst assemblage of olivine and/or clinopyroxene and/or plagioclase (Eq. 1).

$$C^i_{mix} = C^i_{melt}.F_{melt} + C^i_{plg}.F_{plg} + C^i_{cpx}.F_{cpx} + C^i_{olv}.F_{olv} \qquad (1)$$

where $i$ is a given chemical component ($SiO_2$, $Al_2O_3$, $CaO$, $MgO$, $FeO$, $Na_2O$, $K_2O$, $TiO_2$, $MnO$) and

$$F_{melt} + F_{plg} + F_{cpx} + F_{olv} \cong 1 \qquad (2)$$

For each high-Al magma in turn, we completed the following calculation steps. Firstly, 1000 random samples of plagioclase, clinopyroxene, and olivine compositions were drawn, with replacement, from the EPMA data of Volcàn Wolf magmas. Next, for each of the 1000 draws, the carrier melt was approximated by subtracting the solid macrocryst composition, using the macrocryst mass proportions and abundances identified using QEMSCAN maps, from the high-Al magma composition. The median, 5th percentile, and 95th percentile of the calculated carrier melts, which include the 1000 mineral draws and hence propagate the uncertainty of the true mineral compositions for a given sample, were then calculated. Finally, calculated carrier melts were compared with the composition of natural whole-rock data to confirm our hypothesis (Supplementary Fig. 3).

### Quantitative evaluation of minerals by scanning electron microscopy (QEMSCAN)

QEMSCAN maps were collected on a Quanta 650F, field emission gun SEM, equipped with two Bruker XFlash 6130 energy-dispersive X-ray spectrometers in the Department of Earth Sciences at the University of Cambridge. EDS spectra were collected per pixel and combined with a BSE threshold that was calibrated against quartz, gold, and copper standards. The analyses were performed at 25 kV and 10 nA, with a resolution (i.e. pixel size) of 7.5–15 μm$^2$. The software was set to count ~2000 X-ray counts per pixel to aid fast analysis. We montaged all tiles of raw pixel data (1000 by 1000 microns per tile) and exported them as a single CSV data file where each matrix row corresponded to a single pixel and each column to the concentration of a given chemical element in that pixel. The raw QEMSCAN data were subjected to a custom-made post-processing methodology written in R[65]. For each sample, we performed the following steps, summarised below.

The raw QEMSCAN csv file was converted to a stack of 2D chemical maps using the 'raster' package[66] in R. A rectangular area of the thin-section-scale map was cropped to remove the thin-section border whilst retaining the maximum amount of geologically relevant sample.

The stacked chemical map was downsampled from its original resolution by a factor of 3 using a mean filter. As with SEM maps, QEMSCAN maps are notoriously noisy due to their rapid acquisition and relatively low single-pixel count rates. We found that downsampling resulted in a marked reduction in noise and the appearance of coherent mineral zones, which were consistent with BSE imaging, thin-section observations, and EPMA transects. Furthermore, downsampling helped to smooth single-pixel noise within the fine-grained matrix, allowing a coherent matrix group to be identified that was chemically distinct from the macrocrysts (see below). This noise is likely the reason that the proprietary QEMSCAN software, not used in this study, tends to identify hundreds of discrete 'mineral phases' in a single thin section, which necessitates extensive post-processing.

A random forest classifier was applied to the chemical map in order to produce a phase map of plagioclase, clinopyroxene, olivine, and matrix within the sample. The random forest classifier was previously trained on a series of hand-labelled exemplar regions, which were polygonised from 10 random samples mapped by QEMSCAN. The random forest algorithm, deployed via the 'randomForest' package in R[67], used 8 chemical features (Si, Al, Ca, Mg, Fe, Na, K, Ti) for phase prediction, 500 decision trees, and an mtry (number of variables randomly sampled as candidates at each split) of 3.

The crystal size–shape parameters for plagioclase, clinopyroxene, and olivine were then extracted from the phase map. Individual (i.e. entirely surrounded by matrix or different phases) crystals or crystal clots were identified using the 'clump' function in the 'raster' package[66] in R. The 'clump' function gives each individual crystal or crystal clot a unique ID that allows the extraction of its unique size–shape characteristics. Note that,

although splitting touching crystals of the same mineral phase (i.e. mono-mineralic glomerocrysts) into separate grains in chemical maps is notoriously difficult without the use of Electron Backscatter Diffraction (EBSD), solving this problem is of no consequence for our study; we were interested only in separating big crystals (or big monomineralic glomerocrysts) from the fine-grained matrix in order to calculate macrocryst–matrix proportions in a large 2D sample set ($n = 32$). We petrographically identified that big crystals aggregate with other big crystals to make those big crystals bigger, which only helps their identification as being part of the macrocryst load on the basis of their size. In other words, no significant cases of matrix crystals aggregating such that they were misclassified as macrocrysts were found. For each identified crystal (or monomineralic glomerocryst), the length, width, and area were extracted using an equal-area ellipse.

The populations of matrix crystals and macrocrysts were identified using x–y plots of size–shape parameters. For clinopyroxene and olivine, the cut off was largely based on a strong bimodality in crystal area versus crystal number density (i.e. a rare number of very large crystals at odds with the sample set as a whole were ascribed as macrocrysts), whereas for plagioclase the cut off was based on a strong bimodality in aspect ratio versus crystal area (small, high-aspect-ratio matrix crystals versus large, low-aspect-ratio macrocrysts). By using our custom workflow, we were able to select a population threshold and then colour code the QEMSCAN maps in real time for the two crystal populations, allowing textural verification of our size–shape thresholding in 2D. Hence, both 2D data and petrography are informing our definition of a macrocryst rather than some arbitrary, globally applied cutoff. Once selected, matrix crystals were re-assigned to the matrix label in our phase map and were not considered further. There are, of course, limitations to the degree of binary separation that can be achieved between matrix crystals and macrocrysts, which we have attempted to balance.

Phase abundances of macrocrysts and matrix in a given sample were calculated without vesicles using the QEMSCAN-derived phase map (Supplementary Data 4). Area-based calculations are converted to mass-based calculations using average densities for each mineral phase from mean EPMA chemistry (plagioclase = 2.727 g/cm$^3$ [XAn = 0.78]; olivine = 3.469 g/cm$^3$ [XFo = 0.78]; clinopyroxene = 3.168 g/cm$^3$ [Wol = 0.43, Ens = 0.45, Fer = 0.12]). These densities were calculated according to end-member densities (anorthite, albite, forsterite, fayalite, wollastonite, enstatite, ferrosilite)[64].

### Data availability

The data required to interpret, replicate and build upon the methods or findings in this study (Rdata files, CSV files) are available at FigShare https://doi.org/10.6084/m9.figshare.30401920.

### Code availability

The code required to reproduce results is available at FigShare https://doi.org/10.6084/m9.figshare.30401920.

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

## Acknowledgements

Great thanks go to Victoria Smith and Emma Horn for excellent EPMA assistance at the University of Oxford, and Paul Guyett for acquiring exploratory SEM maps at Trinity College Dublin. This work has benefited from useful discussions with Jack Beckwith and Lydia Whittaker. We also acknowledge fieldwork support from the Charles Darwin Foundation, the Galápagos National Park, Sally Gibson, and the crew of MV Pirata. Oliver Higgins would like to gratefully acknowledge the Swiss National Science Foundation for funding through an Early Postdoctoral Mobility Fellowship (Project Number: P500PN_210239). This publication has emanated from research supported in part by a research grant from Science Foundation Ireland and the Geological Survey of Ireland under the SFI Frontiers for the Future Programme 20/FFP-P/8895, and a Charles Darwin and Galápagos Islands Junior Research Fellowship at Christ's College, Cambridge (awarded to M.J.S.). Additional fieldwork funding was provided by the Jeremy Willson Charitable Trust (administered by the Geological Society of London) and the Mineralogical Society of Great Britain and Ireland. David Neave acknowledges contributions from a NERC grant (NE/T011106/1). Dennis Geist's work was supported by the U.S. National Science Foundation.

## Author contributions

O.H. drafted the main text and figures, interpreted the data, performed analyses, and wrote the code. M.S. acquired funding for fieldwork, collected samples, performed analyses, assisted with data interpretation, and provided supervision. D.G. and D.N. assisted with data interpretation. I.B. performed the QEMSCAN analyses and advised on data treatment. B.B. and M.G. contributed to fieldwork and sample collection. All authors provided feedback for the final draft.

## Competing interests

The authors declare no competing interests.
