## [Transparent Peer Review file · Communications Earth & Environment]

Annual-to-millennial fluctuations in the physical properties of crystal-rich magma storage zones

Corresponding Author: Dr Oliver Higgins

This manuscript has been previously reviewed at another journal. This document only contains information relating to versions considered at Communications Earth & Environment.

Version 0:

Decision Letter:

Dear Dr Higgins,

Your revised manuscript titled "Annual-to-millennial fluctuations in the physical properties of crystal-rich magma storage zones" has now been seen by the original reviewers 1 and 2, whose comments appear below. In light of their advice we are delighted to say that we are happy, in principle, to publish a suitably revised version in Communications Earth & Environment.

We therefore invite you to revise your paper one last time to address the remaining concerns of our reviewer 2. At the same time we ask that you edit your manuscript to comply with our format requirements and to maximise the accessibility and therefore the impact of your work.

EDITORIAL REQUESTS:

******Please take care to match our formatting and policy requirements. We will check revised manuscript and return manuscripts that do not comply. Such requests will lead to delays. ******

SUBMISSION INFORMATION:

OPEN ACCESS:

Communications Earth & Environment is a fully open access journal. Articles are made freely accessible on publication. For further information about article processing charges, open access funding, and advice and support from Nature Portfolio, please visit <https://www.nature.com/commsenv/open-access>

Link Redacted

**This url links to your confidential home page and associated information about manuscripts you may have submitted or be reviewing for us. If you wish to forward this email to co-authors, please delete the link to your homepage first **

Best regards,

Alireza Bahadori, PhD
Associate Editor
Communications Earth & Environment
Consulting Editor
Communications Sustainability

REVIEWERS' COMMENTS:

Reviewer #1 (Remarks to the Author):

My comments have been address and I do not have further suggestions. Congratulation to the authors for the great work.

Reviewer #2 (Remarks to the Author):

Overall, I find this manuscript a valuable contribution to the igneous petrological literature and suitable for publication in Nature Communications Earth & Environment. I have only a few minor comments:

1) L144f & response to my comment on L152/153 in the first submission – and use of the term “liquid line of descent” generally: In an equilibrium crystallisation scenario where no minerals are fractionated, we would not expect to see *any* compositional trends in the whole rock. The liquid line of descent should be evident in glass and mineral compositions, not in whole-rock compositions. Hence, all magmatic glasses should fall along the low-Al liquid line of descent (which they seem to do). However, we would not expect to see this trend in *whole-rock* compositions (unless the magma actually lost plagioclase) – yet the sentence in L144 implies otherwise. I suggest rephrasing the sentence to clarify this and avoid inaccurate use of the term.

2) L149: “(iv) no relationship between plagioclase macrocryst composition and whole-rock Al.”: This point isn’t clearly linked to the preceding discussion, unlike the other three points. Please reference the relevant section for consistency.

3) L343: typo: “macorcryst”

** Visit Nature Portfolio's author and referees' website at www.nature.com/authors for information about policies, services and author benefits **

Reviewer #1 (Remarks to the Author):

My comments have been address and I do not have further suggestions.
Congratulation to the authors for the great work.

Many thanks for your previous comments. They helped to improve the manuscript.

Reviewer #2 (Remarks to the Author):

Overall, I find this manuscript a valuable contribution to the igneous petrological literature and suitable for publication in Nature Communications Earth & Environment. I have only a few minor comments:

1) L144f & response to my comment on L152/153 in the first submission – and use of the term “liquid line of descent” generally: In an equilibrium crystallisation scenario where no minerals are fractionated, we would not expect to see *any* compositional trends in the whole rock. The liquid line of descent should be evident in glass and mineral compositions, not in whole-rock compositions. Hence, all magmatic glasses should fall along the low-Al liquid line of descent (which they seem to do). However, we would not expect to see this trend in *whole-rock* compositions (unless the magma actually lost plagioclase) – yet the sentence in L144 implies otherwise. I suggest rephrasing the sentence to clarify this and avoid inaccurate use of the term.

Matrix glasses and low-Al whole rock compositions are chemically very similar (if not identical), something which we discuss with reference to Fig. 1a. The main difference is cooling history (one has nucleated matrix crystals, the other not). But chemically they overlap in all elements and so can likely be considered a liquid/melt.

2) L149: “(iv) no relationship between plagioclase macrocryst composition and whole-rock Al.”: This point isn’t clearly linked to the preceding discussion, unlike the other three points. Please reference the relevant section for consistency.

Added link to supplementary figure 4

3) L343: typo: “macorcryst”

Amended